# Interprofessional collaboration during a specialised mobile palliative care service pilot in the rural area of Lucerne

**Sahra Maria Anna Bucher** [1]*, **Anne Marie Schumacher Dimech**[1], **Beat Müller**[1,2], **Patrick E. Beeler**[3]

1 Faculty of Health Sciences and Medicine, University of Luzern, Luzern, Switzerland, 2 Cantonal Hospital Luzern, Palliative Care, Luzern, Switzerland, 3 Center for Primary and Community Care, Faculty of Health Sciences and Medicine, University of Lucerne, Lucerne, Switzerland

* sahra.bucher@hotmail.com

**Data Availability Statement:** All relevant data are within the manuscript and its Supporting Information files.

**Funding:** The author(s) received no specific funding for this work.

## Abstract

Interprofessional collaboration in outpatient palliative care is critical to ensuring good quality of care in the home care sector. We investigated facilitators and barriers (FaBs) of interprofessional collaboration among healthcare professionals who participated in a 6-month pilot of a newly implemented specialised mobile palliative care service (SMPCS) in rural Lucerne. This study used a mixed-methods approach to collect (i) qualitative data on FaBs as perceived by nurses and primary care physicians (PCPs), and (ii) quantitative data across the entire interprofessional collaboration using a validated questionnaire expanded with 10 specific questions about the pilot. Identified facilitators of interprofessional collaboration were (i) use of standardised documents, (ii) clear allocation of responsibilities, (iii) regular exchange and clear communication and (iv) consideration of care coordination. Reported barriers were (i) a deficit of knowledge and experience of palliative care among PCPs and (ii) time constraints. This study provides valuable insights into FaBs of interprofessional collaboration in palliative care. Several recommendations can be drawn for how interprofessional collaboration may be optimised. Awareness of FaBs and their consideration in the implementation phase of new services can strengthen the foundation for a successful interprofessional collaboration.

## 1 Introduction

Interprofessional collaboration exists when "multiple health care workers from different professional backgrounds work together with patients, families, caregivers, and communities to provide quality care" [1]. In the outpatient setting, interprofessional collaboration can improve patient satisfaction, acceptance of care and health outcomes. As a result, interprofessional collaboration enables health professionals to work at their highest capacity and to address complex health situations in the best possible ways [1]. In Switzerland, the decentralisation of various professional groups demands effective interprofessional collaboration in the outpatient palliative care setting. Unfortunately, a lack of communication and responsibility is observed

**Competing interests:** The authors have declared that no competing interests exist.

more frequently in outpatient settings in rural areas than in the well-connected urban areas. Such shortcomings in interprofessional collaboration can lead to the inadequate treatment of patients [2]. To optimize interprofessional collaboration, its specific facilitators and barriers (FaBs) must be identified [3]. During a pilot phase, several professional groups often come together for the first time, making interprofessional collaboration even more challenging [4]. So far, there is no Swiss study of interprofessional collaboration and its FaBs as part of a pilot in the palliative care outpatient setting. Due to this lack of data, this study aimed to glean further knowledge with focus on the specialised mobile palliative care service (SMPCS) project in rural Lucerne. SMPCS is a novel outpatient service and its pilot was conducted from August 1 to December 31, 2022 [5, 6]. The objective of this study was to identify FaBs of interprofessional collaboration during the SMPCS pilot in rural Lucerne in Switzerland. The following research question was addressed: what were FaBs of interprofessional collaboration during the SMPCS pilot project in a rural part of Switzerland?

## 2 Methods

### 2.1 Study design

This study uses a mixed methods approach to (i) develop qualitative insight into the FaBs of interprofessional collaboration from the point of view of home care nurses and primary care physicians (PCPs); and (ii) to collect quantitative data from all professional groups who had direct or indirect patient contact as part of the SMPCS pilot. This study defines professional groups as a collection of diverse professionals who have direct or indirect interactions with patients receiving palliative care. The primary group within the specialised palliative care service consisted of qualified nurses with further training in specialised palliative care and primary care physicians. The nursing professionals were employed within existing home care organizations. The pilot project was structured in a highly dynamic manner whereas additional professional groups were incorporated based on the patients' specific needs. This approach facilitated individualized, specialised, and patient-centered care for each palliative care patient. There were no predefined individuals assigned to the pilot project from the outset; rather, they were integrated into the pilot project team as needed throughout its course. The use of mixed methods provided the opportunity to gain a deeper understanding of the interprofessional collaboration's FaBs through interviews and to include the opinions of further involved professional groups via a questionnaire. Our study adhered to the consolidated criteria for reporting qualitative research (COREQ) guideline and checklist for reporting of survey studies (CROSS) [7, 8].

### 2.2 Participants

Nurses of the home care organisation providing SMPCS were approached via email and telephone with an invitation to participate in this study. In this first contact, potential participants were informed about the study and its objective. Participation was voluntary throughout the study.

**2.2.1 Qualitative part.** Interview participants were recruited in person, by telephone and by email. Only qualified nurses and PCPs, aged 18 and above, who had direct or indirect contact with patients at least once in the context of the SMPCS pilot project were approached and included for the interviews. Interviews were conducted on a one-off basis. Three female nurses (aged 42, 46 and 41) were interviewed, all of whom had further training in palliative care and two of whom had a specialised responsibility. One female PCP (aged 44) and two male PCPs (aged 32 and 37) without further training in palliative care were interviewed. All nurses who were approached agreed to be interviewed. Six PCPs were contacted, only three agreed to be

interviewed. Contact with the involved PCPs was established through the nurses. All interviewees were unknown to the interviewer.

**2.2.2 Quantitative part.**   Professionals involved in the SMPCS pilot were invited by email to participate in the online interprofessional collaboration questionnaire. Only individuals aged 18 and older who had direct or indirect contact with patients were included. A total of 39 professionals from seven different professional groups were invited to participate. All professionals were categorized into one of the following seven professional groups: primary care physicians, nurses, psychologists, psychotherapists, medical practice assistants or specialised medical assistants, occupational therapists, nutritionists, social workers, pastoral care providers, and volunteers. Through the nurses, contact was established with other professional groups who had direct contact with SMPCS-involved patients or were heavily involved in their cases.

### 2.3 Data collection

**2.3.1 Qualitative data collection.**   Between August 1 and December 31, 2022, six semi-structured interviews were conducted in German, the regional language, by a single interviewer, without further training in conducting interviews. The interviews lasted between 18 and 22 minutes with PCPs and 59 to 72 minutes with nurses. Four interviews were conducted in person at the respondents' workplaces and two via telephone. Before conducting the interviews, explanations were provided about the aim of the study, the reasons for recording the interviews and data confidentiality. Participants provided informed consent based on the Swiss federal law on data protection and were informed that they would remain anonymous. The anonymization procedure was adopted from Kuckartz & Rädiker (2022) and applied according to their guidelines [9]. As no patient data were involved, an ethics committee application was not required under the Swiss federal act on research involving human beings [9, 10]. All recorded interviews were deleted after transcription. The semi-structured interviews were based on interview guides for nurses and PCPs, which was forwarded to the second author for proofreading and was pilot tested during two mock interviews to strengthen its validity. The interview guides addressed communication, exchange, role distribution, organisational processes and demographic data.

**2.3.2. Quantitative data collection.**   The German-language online questionnaire about interprofessional collaboration was sent at the end of the pilot between December 20, 2022 and January 10, 2023. In each case, a reminder letter was sent two weeks after the first request. Before sending out the questionnaire, it was reviewed and edited by five different researchers to strengthen its validity. The online questionnaire was conducted anonymously via Google Forms, with access to the data always protected. The emails sent and the response rate was compared to prevent multiple participation. It contained 13 standardised questions of a validated questionnaire about interprofessional collaboration [11], which was expanded with ten specific questions about the pilot: one question addressed the importance of interprofessional collaboration, three questions addressed advance directives, two questions addressed healthcare proxy, one question addressed the accessibility of PCPs and three questions addressed SMPCS standardised documents. Non-response was not addressed, item weighting was not conducted and a sample size calculation was not performed, due to the goal of describing the small group of involved professionals.

### 2.4. Data analysis

**2.4.1 Qualitative data analysis.**   Interviews were transcribed using the program f4transkript V8.1.1 (811) (audiotranskription, Marburg, Germany). The interviewer transcribed all

six interviews according to predefined transcription rules [12, 13]. After transcription, content analysis was performed using the program f4analysis 3.4.2 (audiotranskription, Marburg, Germany). A deductive content analysis was primarily conducted, using a predefined coding manual. We aimed to construct our work upon pre-existing, evidence-based knowledge, utilizing Reeves' theoretical framework, which delved into interprofessional collaboration and delineated four primary categories within its construct, namely: relational, procedural, organizational, and contextual [14]. This study primarily adopted a deductive approach as its main strategy for three reasons. (I) A deductive approach was chosen because the utilization of the theoretical framework provides a solid foundation for data analysis derived from relevant theories and existing knowledge. (II) A deductive approach can offer the opportunity to focus the analysis on specific aspects that are relevant to the research question. (III) This approach facilitates the systematic categorization of the interview content according to the predetermined framework and its concepts, enhancing the reliability and validity of the analysis. Codes have been added to the manual and noted as such. All six interviews included the application of these four categories of Reeves' theoretical framework. In terms of content, the interviews with nursing staff were fully consistent with the aforementioned categories. In contrast, the interviews with primary care providers only matched two of the four defined categories, namely: relational and procedural. To assure inter-coder reliability, the interviewer (first author) and the second author both conducted the content analysis based on the predefined coding handbook. Coding discrepancies between the two researchers were discussed, recorded and resolved. Inter-coder-reliability was 81.2% and was calculated using the program MAXQDA Plus 2022 22.4.1 (MAXQDA, Berlin, Deutschland). The transcripts and content analysis were not presented to the participants for feedback.

**2.4.2 Quantitative data analysis.** Descriptive data analysis was performed using STATA 17.0 (StataCorp, College Station, TX, USA). Categorical variables are presented as counts, continuous variables with non-normal distributions as medians with first and third quartiles. Graphs were generated using Microsoft Excel 16.66.1 (Microsoft, 2022).

## 3 Results

### 3.1 Qualitative part

The qualitative section highlights seven key findings from the analysis, relying on two primary factors from Reeves' theory: processual and relational. Processual factors, detailed in chapters 3.1.1 to 3.1.4, encompass elements like time, routines, technologies, and situational complexity, influencing interprofessional collaboration. Similarly, relational factors, discussed in chapters 3.1.5 to 3.1.7, include socialization, team dynamics, roles, and professional authority, contributing to interprofessional collaboration [14].

**3.1.1 Evaluation of interprofessional collaboration and professionals involved.**
According to the PCPs, interprofessional collaboration mainly involved work with nurses, with one PCP stating that he also had contact with physiotherapists. Nurses stated that they had contact with various professional groups, including pastoral carers, physiotherapists, psychologists and other service providers, such as health insurance, material suppliers or the pharmacy. All interviewees reported that they knew the term interprofessional collaboration. Interprofessional collaboration was important for PCPs due to the shortage of PCPs, the specialization of various professions in recent decades and the changed role of PCPs. For instance, one PCP explained, "I as a doctor couldn't do it all, it's not like 60 years ago". From the nurses' perspective, interprofessional collaboration is necessary due to the high complexity of the patients, which requires knowledge from different professional groups. One nurse stated that

interprofessional collaboration is more demanding in the outpatient setting due to the decentralisation of various professional groups that do not work in the same institution.

**3.1.2 Standardised documents in specialised palliative care.** All participants interviewed were satisfied with the SMPCS-specific standardised documents and with the fact that these were digitally available. Two nurses stated that the lack of a shared electronic tool hinders interprofessional collaboration. Despite the initial extra work, the PCPs and nurses described the documents as making their work more efficient. Both said the documents would pay off in the long term by increasing nurses' independence and reducing exchanges between PCPs and nurses. One PCP stated that when filling in the standardised forms "it's all at once, maybe [not all medications are necessary], but then they have everything right from the start. In the end, that's probably more efficient, yes".

Further standardised forms, such as the advance directive, are considered an essential element of palliative care by two PCPs and three nurses, however, one PCP does not attach much importance to this document. PCPs do not usually ask about an advance directive. One PCP stated that the nursing staff informs the PCPs if a patient does not have an advance directive, and nursing staff may also discuss the advance directive with a patient and forward it to the PCP for filing. One nurse stated that she was not sufficiently trained to adequately educate and advise a patient in completing their advance directive. According to the nursing staff, the advance directive is often not available, but resuscitation status is always clearly indicated.

**3.1.3 Time factor in interdisciplinary teams.** Due to the high workload and limited time available, correspondence via email and telephone is demanding for the PCPs. PCPs see correspondence as additional work they would like to minimize. For urgent concerns, all PCPs agree to take their time and deal with it as quickly as possible, which is confirmed by nurses. According to all nurses, the preparation of care plans for patients in the SMPCS takes more time than for regular patients because of the high complexity of these patients. Complex patient situations can suddenly change and require quick action. When professional groups are difficult to reach or it is unclear who to approach, this may hinder the involvement of other professional groups. One nurse stated that for such involvement, "you have to make phone calls [. . .] and that takes so much time [. . .], [which] is then lost on patient care".

**3.1.4 Care coordination in specialised palliative care.** Two PCPs and all the nurses felt that improved care coordination between involved professional groups would facilitate their work and interprofessional collaboration. According to all interviewees, care coordination would make sense for complex cases or cases involving several professional groups, but if mainly nurses and PCPs were involved, such a role would not be necessary. One PCP mentioned that he saw his professional group as the most suitable for care coordination, since requests from various professional groups often require a prescription by the PCP anyway. On the other hand, the PCP stated that "you also have to see that the PCP doesn't always have time for all professional groups to make long phone calls or answer emails". Another PCP and two nurses address the role to nursing, whereas one nurse stated that the relationship between the professional group and the patient goes far beyond care coordination. Further, two nurses with extended responsibilities have unofficially taken on a large part of the care coordination, but none of them had any further training, nor were they compensated for doing so or specifically hired for this role.

**3.1.5 Team roles in interdisciplinary teams.** All participants interviewed considered it essential to clarify professional roles and their responsibilities to avoid misunderstandings or over-/underestimation of competencies. According to the nurses, a clear understanding of roles requires that the professional groups exchange information, know each other and know professional groups' contributions to patient care. At the same time, a clear distribution of roles supports effective and efficient interprofessional collaboration. One nurse stated "that you know what everyone's role is in this team, I think that's the most important thing".

**3.1.6 Knowledge and experience in specialised palliative care.** All three nurses stated that a basic understanding of palliative care must be present for successful interprofessional collaboration in the context of the pilot. Without knowledge of palliative care, it is challenging to work with other professional groups. The nurses cited the lack of PCPs' knowledge about important palliative care medications, dosages or combinations of medications. In particular, recognising and accepting palliative care, without initiating measures beyond addressing quality of life, is an important issue from the nurses' perspective. One nurse stated that "someone who already has some experience as a [PCP], I notice [. . .] that maybe they don't do anything more in terms of therapy, but only strive for the best possible symptom relief and quality of life".

**3.1.7 Exchange and communication in interdisciplinary teams.** All interviewees prefer face-to-face exchanges, although meeting face-to-face is often not possible due to high workload, so email and telephone are considered sufficient means of communication. Digital exchange platforms such as Zoom or Teams are conceivable among nurses, but not preferred. No interprofessional meetings were held during the pilot, however regular meetings were held between the nurses, which, according to them, was very helpful for their work in palliative care. Overall, all interviewees described exchange and communication as satisfactory and an essential part of interprofessional collaboration. One PCP stated that "I really think good communication is the be-all and end-all between different professionals". According to the interviewees, integrating suggestions from other professional groups is beneficial for interprofessional collaboration.

## 3.2. Quantitative part

In total, 17 (43.6%) out of 39 respondents completed the questionnaire providing further information about their experience with interprofessional collaboration during the pilot. All interviewed nurses participated in the questionnaire, and two out of three PCPs interviewed. The demographic data of the questionnaire respondents are displayed in Table 1.

**3.2.1 Evaluation of interprofessional collaboration and professionals involved.** All questionnaire participants ($n = 17$) indicated that they viewed interprofessional collaboration as important and meaningful. Fig 1 shows how often the professional groups reported having been in contact with other professional groups during the pilot.

Participants stated to have the most contact on a daily base with PCPs ($n = 6$), medical practice assistants or specialised medical assistants ($n = 7$) and nurses ($n = 7$). Fig 2 shows how many professionals agreed with the different statements about other professional groups with whom they work at least once a month.

Professionals from the professional group nurses ($n = 3$) and PCPs ($n = 1$) agreed to the statement "disagreements with the other professional group often go unresolved". Professional groups of medical practice assistants or specialised medical assistants ($n = 3$), nurses ($n = 3$), PCPs ($n = 2$) as well as the volunteer ($n = 1$), specialised physician ($n = 1$) and psychotherapist ($n = 1$) disagreed on the statement. To two professionals from the professional groups nurse and PCP did the statement not apply.

**3.2.2 Standardised documents in specialised palliative care.** The questionnaire included three statements about each of the three standardised documents. Responses to these statements are displayed in Fig 3.

The three standardised documents were considered necessary by most professional groups. The three statements of necessity did not apply to two professionals from the medical practice assistant or specialised medical assistant group and specialised physician group. Most professionals agreed with the statements about the completeness of all three standardised documents, except for two professionals from the professional group medical practice assistant or

**Table 1. Demographic data of quantitative part.**

| Demographic data of quantitative part | n |
|---|---|
| *Professional groups* | |
| Nurse | 7 |
| PCP | 4 |
| Medical practice assistant or specialised medical assistant | 3 |
| Psychotherapist | 1 |
| Volunteer | 1 |
| Specialised physician in palliative care | 1 |
| *Gender* | |
| Female | 14 |
| Male | 3 |
| *Age (years)* | |
| Q1 | 41 |
| Median | 44 |
| Q3 | 47 |
| *Employment* | |
| Employed | 16 |
| Self-employed/own company | 1 |
| *Workload (%)* | |
| Q1 | 40 |
| Median | 60 |
| Q3 | 90 |
| *Hours worked per week (hours)* | |
| Q1 | 20 |
| Median | 26 |
| Q3 | 42 |
| *Professional work experience (years)* | |
| Q1 | 10 |
| Median | 16 |
| Q3 | 20 |
| *Further education in PC* | |
| Nurses | 7 |
| Specialised physician in palliative care | 1 |

Abbreviations: primary care physicians (PCP), quartile 1 (Q1), quartile 3 (Q3).

specialised medical assistant. To six professionals from the professional groups nurse ($n = 2$), specialised physician ($n = 1$), PCP ($n = 1$) and medical practice assistant or specialised medical assistant ($n = 2$), the three statements of completeness did not apply. Fig 4 shows the results of the statements about the advance directive and the healthcare proxy.

The advanced directive was seen by all professional groups as necessary, except for two professionals of the professional groups nurse ($n = 1$) and PCP ($n = 1$), who disagreed with the first statement.

**3.2.3 Time factor in interdisciplinary teams.** The questionnaire included one question about the accessibility of the PCPs. To the statement "the responsible PCP can be reached by telephone in case of ambiguities and responds promptly", professional groups either strongly agreed ($n = 5$), agreed ($n = 9$), strongly disagreed ($n = 1$), or found the statement not applicable ($n = 2$).

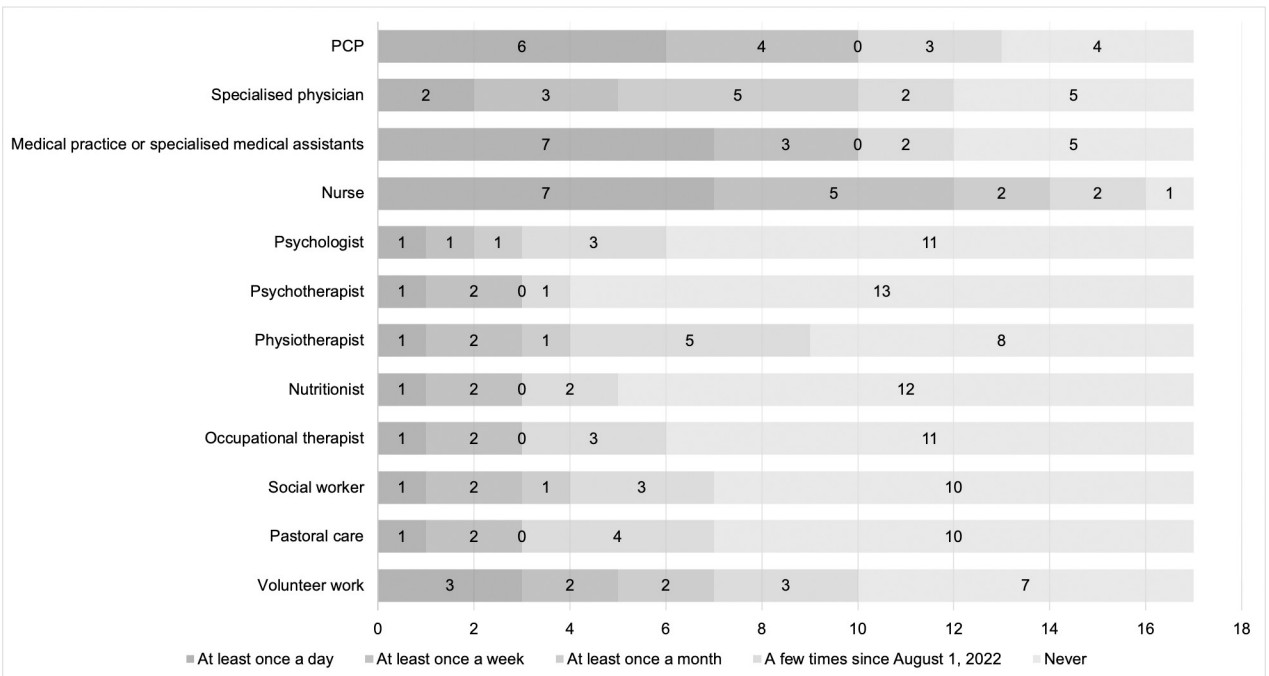

**Fig 1. Frequency of interprofessional collaboration between August 1 and December 31, 2022.**

**3.2.4 Team roles in interdisciplinary teams.** Fig 5 shows how professionals agree with the other listed professional groups on their responsibilities. Responsibilities were assessed if cooperation between the professional groups took place at least once a month.

## 4 Discussion

This study found relational and processual factors to be relevant FaBs [14]. Facilitators were (i) the use of standardised documents, (ii) a clear allocation of responsibilities to the professional groups involved, (iii) the consideration of care coordination, and (iv) clear communication and exchange between professional groups. Barriers were (i) time constraints and (ii) differences in knowledge and experience among professional groups.

### 4.1 Evaluation of interprofessional collaboration and professionals involved

All interviewees and most questionnaire respondents were satisfied with the interprofessional collaboration. The reason for respondents indicating that statements were "not relevant" may be that they had not encountered the situations in question and therefore did not have a reference point. Regarding Fig 2, the frequent collaboration between PCPs, nurses and medical practice assistant or specialised medical assistant showed higher interprofessional collaboration involvement among these groups than among other professional groups. This is unsurprising given that most palliative home care requires the involvement of PCPs, nurses, and medical practice assistants or specialised medical assistants [15, 16]. The reasons for the low involvement of occupational therapists, pastors, social workers, nutritionists, and psychologists may be explained by the new pilot set-up, time constraints, high administrative complexity, or simply the perceived lack of need for other professional groups to be involved. Such networking with other professional groups, known as integrated care, is nowadays increasingly

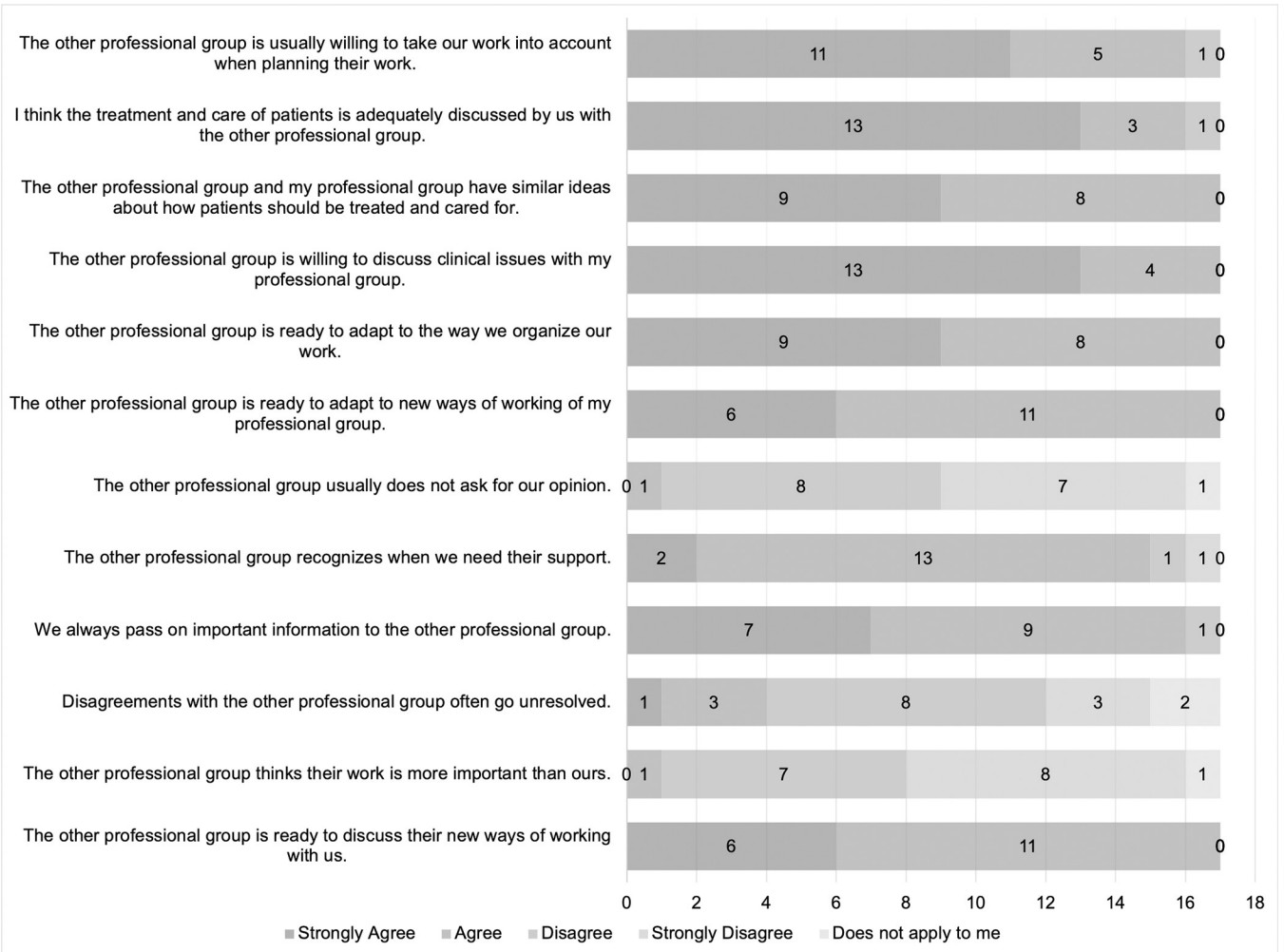

**Fig 2. Evaluation of interprofessional collaboration.**

promoted to meet the needs of demographic and epidemiological changes [17]. Through the further development of the structures, processes and services of the outpatient and inpatient healthcare systems, better coordinated service can be provided to align care more closely with patients' wishes and use resources more efficiently [17].

## 4.2 Facilitators of interprofessional collaboration

During the pilot project, four key facilitators of interprofessional collaboration emerged. Professionals found standardized documents beneficial. Additionally, clear role distribution within the team and regular communication were highlighted. A care coordinator was also suggested to assist in managing complex and heavy workloads.

**4.2.1 Standardised documents in specialised palliative care.** The three standardised documents used were developed by several professional groups and were made publicly available by the umbrella organisation "Palliativ Luzern". These documents proved crucial in providing common ground for nurses and PCPs. The interview outcomes and the questionnaire results showed that most of the professional groups involved considered these documents necessary. Nevertheless, the three statements on completeness did not apply to six respondents in the

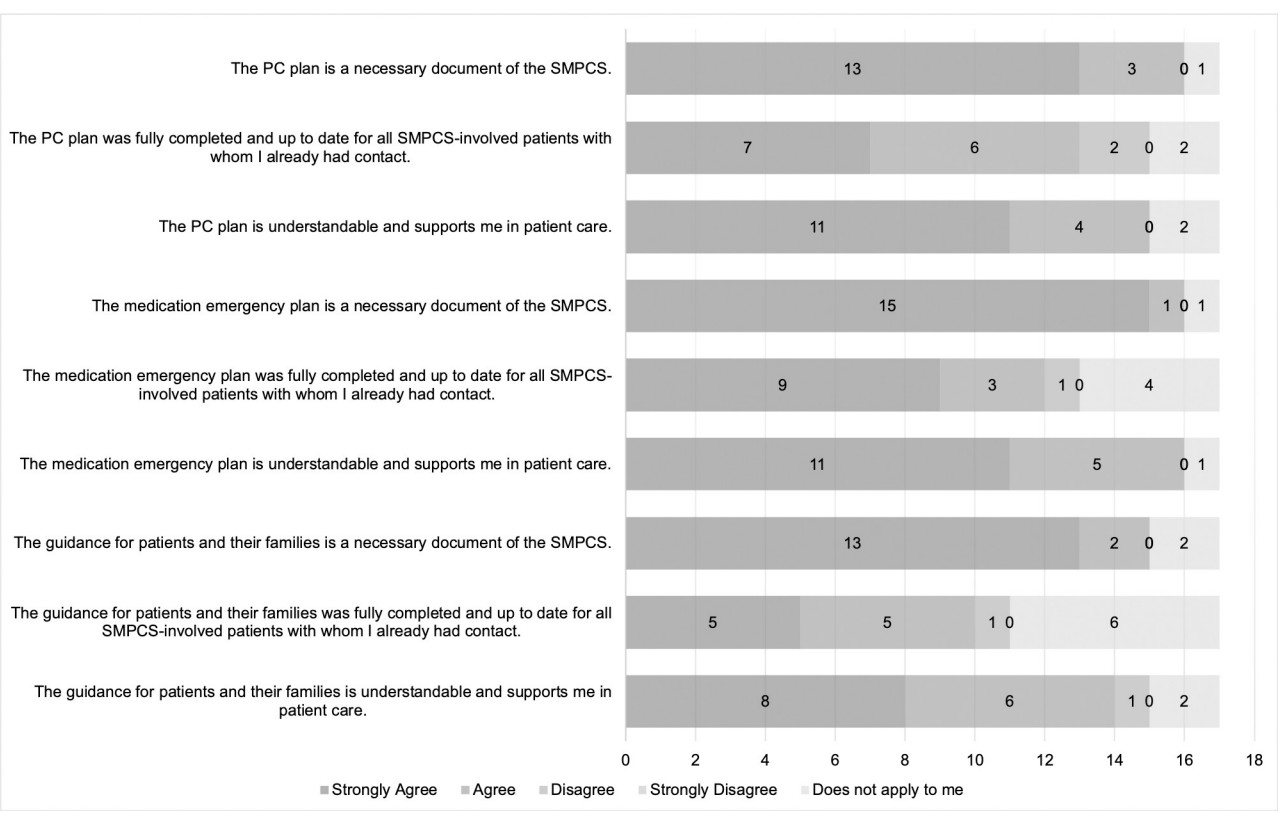

**Fig 3. SMPCS-specific standardised documents.**

questionnaire. The reasons for individuals stating that they were not familiar with the documents may be associated with their professional groups and competencies. Other studies confirm reduced information loss and higher collaboration within and between professional groups as the advantages of standardised documents [18, 19]. The literature also shows that a

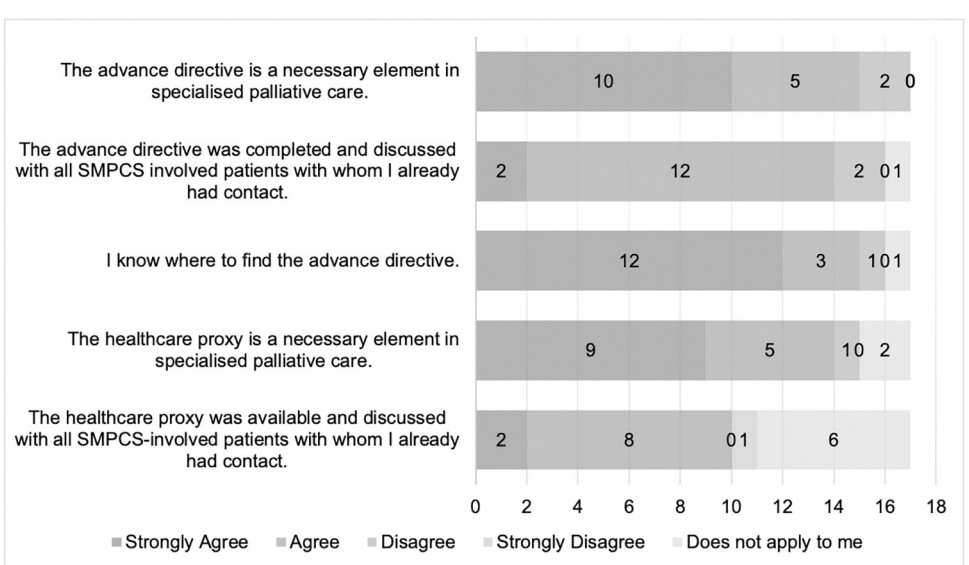

**Fig 4. Further standardised documents.**

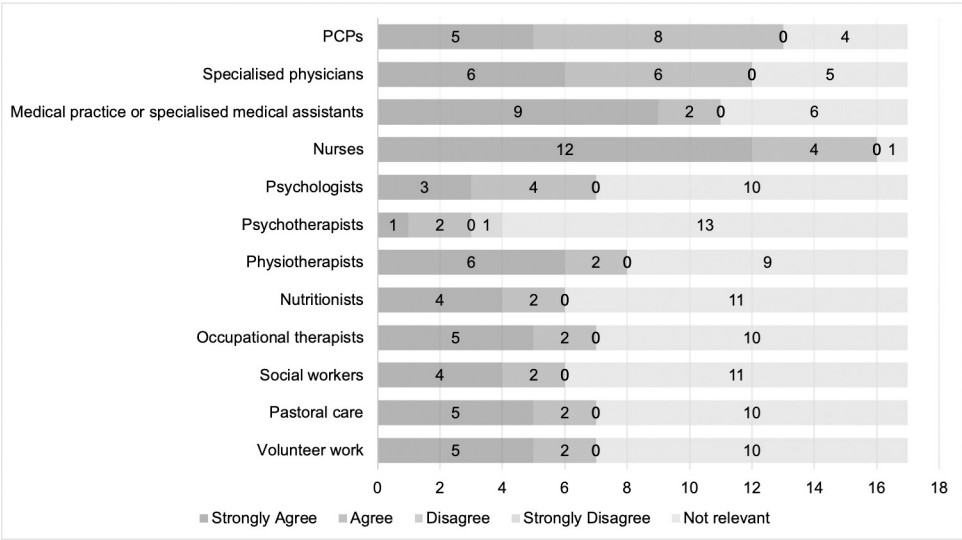

**Fig 5. Agreement on responsibilities.**

shared e-tool for exchanging documents can be beneficial for interprofessional collaboration and that the general electronic transfer of documents and information is recommended [2, 20]. PCPs further pointed out the preparatory work that is required when using standardised documents, which is known in literature as advance care planning [21, 22]. In advance care planning, documents are prepared in advance to allow users to act independently and quickly in emergencies. Advance care planning helps prevent hospitalizations, enables nurses to work more independently, and reduces unnecessary contact with professional groups [21, 22].

The results from the questionnaire indicated that the advance directive and the healthcare proxy were considered necessary documents for palliative care. Two respondents who did not share this opinion were from the PCP and nursing professional groups. Reasons for this discrepancy among PCPs and nurses were not known. The lesser importance ascribed to these documents by the nurse and PCP–professional groups who are most involved with patients–shows possible barriers to the use of both documents. In the everyday interaction between nurses and PCPs, the responsibility for completing such documents was unclear, and both professional groups were insufficiently trained for this task. Lack of knowledge and time may be additional reasons for neglecting tasks such as completing and discussing the advance directive with patients [19].

**4.2.2 Team roles in interdisciplinary teams.** As already confirmed by several Swiss studies [16, 19, 23], well-defined roles and responsibilities are a key element of effective interprofessional collaboration. Additionally, the clear distribution of responsibilities can help to decrease the workload by allowing all professional groups to fully contribute their knowledge and experience [16, 19]. Respondents agreed more on their responsibilities with nurses rather than with PCPs. These results may be attributable to the closer involvement of professional groups with nurses than PCPs.

**4.2.3 Exchange and communication in interdisciplinary teams.** During this pilot, interdisciplinary meetings among nurses were frequently held, whereas information about meetings among PCPs regarding palliative care was not available and interprofessional team meetings were not held. Regular and structured interprofessional meetings are described in the literature as a component of successful interprofessional collaboration and a countermeasure to misunderstandings regarding responsibilities and missing role allocations [19].

For PCPs, the exchange with other professional groups was challenging because of PCPs' sizeable workload and the associated need to keep exchanges as short as possible. Nurses indicated the need for respectful and trusting communication to maintain good working relationships and achieve desired results from other professional groups. Studies show that communication is also important to maintain quality of care because good communication reduces information loss and improves work satisfaction [3, 19]. Effective communication is critical for providing information on roles and responsibilities [3, 16].

**4.2.4 Care coordination in specialised palliative care.** A care coordination (or case) manager role has not yet been established in Switzerland in the outpatient setting [3, 16], although care coordinators can assume responsibility for specific tasks–especially those associated with interprofessional collaboration. As described in our study, nurses frequently assume responsibility for care coordination tasks although they are not trained, employed or remunerated for these tasks [16, 24]. The resulting work overload and limited competence can lead to work-related frustration, which can affect interprofessional collaboration. This can be prevented with appropriate training and resources [24]. Examples outside Switzerland show that such care coordination including a member of staff with this specific task has been used for several years as a possible solution to the shortage of PCPs [25]. Nevertheless, most care coordinators are available only in acute settings [26, 27]. Care institutions with experience in employing care coordinators describe coordinators as providing support for everyday work for complex patient situations and report reduced hospitalizations in the last 30 days of life compared with interprofessional collaboration care without care coordinators [28, 29].

## 4.3 Barriers of interprofessional collaboration

The main obstacles cited for interprofessional collaboration during the pilot project were the prevalent time constraints in healthcare and the discrepancies in expertise and experience in specialised palliative care, particularly among PCPs.

**4.3.1 Time constraints in interdisciplinary teams.** Given already tight time constraints, PCPs stated the need to keep additional work to a minimum. In urgent patient cases, priority and time were always allocated to process a case. Lack of time appears to have an impact on the involvement of other professional groups in nursing work. The barriers to involving other professional groups, whose accessibility is more complex, force nurses to prioritise when they experience time constraints and relevant professional groups may not be involved if barriers are too high. As confirmed in the literature, barriers (such as time constraints) and additional work (such as obtaining prescriptions) make involving additional professional groups challenging [19, 23].

**4.3.2 Discrepancies in knowledge and experience in specialised palliative care.** In contrast to nurses, primary care physicians (PCPs) indicated a lack of additional training in palliative care. As supported by another study, only 14% of PCPs had received formal training in palliative care, despite 80%–82% of them being involved in the care of patients at the end of life, despite knowing that specialised palliative care has been shown to enhance the quality of life [30, 31]. Raising awareness of palliative care in medical training influences PCPs' later knowledge and practice of palliative care [30]. Furthermore, interprofessional education in healthcare is an indispensable element of study programmes and impacts future interprofessional collaboration in the work setting [32]. Therefore, further training opportunities have been developed at the University of Lucerne, where a newly created certificate of advanced studies (CAS) in palliative care has been available since 2021 [33–35]. Attendance to such training opportunities remains challenging for PCPs because of competing educational opportunities and the time limits imposed on PCPs for obtaining further education [18, 23].

### 4.4 Strengths and limitations

This study provides a unique and in-depth picture of the FaBs of an SMPCS in rural Lucerne. To the best of our knowledge, no other studies present the FaBs of an SMPCS pilot in Switzerland. This valuable information can be beneficial to future similar projects in rural areas, especially to foster improved interprofessional collaboration.

The limitations of this study should be considered in interpreting the results. We investigated interprofessional collaboration during one pilot, the results are based on six interviews and a small sample of questionnaire responses. Only three of the six PCPs contacted agreed to participate in interviews. The reasons for low interviewee recruitment are unknown but might be associated with a lack of interest, time, or affinity with palliative care. Furthermore, interviewees' contact with only a few palliative care patients may have influenced the interviews. The person conducting the interview had basic knowledge in interviewing, which may have influenced the results. Finally, only 17 people participated in the questionnaire; some professional groups were not suitably represented, e.g. only one specialised physician. Therefore, generalizability may be limited. Nevertheless, several of our findings are in line with the available literature.

## 5 Conclusion

This study gives first valuable insights into FaBs in respect of interprofessional collaboration during a SMPCS pilot in rural Lucerne. Several recommendations became apparent that are necessary for successful interprofessional collaboration. Developing clearly defined roles and responsibilities and holding regular and structured interprofessional meetings should be considered within an interprofessional team. Advance care planning helps to facilitate interprofessional collaboration, as well an effective communication to reduce information loss and improve work satisfaction. Specific roles, such as care coordinators involving other professions may be a solution for improved interprofessional collaboration in the future. Furthermore, interprofessional education in healthcare should be given greater importance, as well as early sensitisation for palliative care in medical training. FaBs should be kept in mind when establishing new interprofessional health teams to avoid retroactively optimising teams, since awareness and consideration of FaBs in the implementation phase of a new SMPCS can provide the basis for successful interprofessional collaboration.

## Supporting information

**S1 File. Questionnaire interprofessional collaboration.**
(DOCX)

**S2 File. SMPCS standardised documents.**
(DOCX)

**S3 File. Interview guide.**
(DOCX)

## Author Contributions

**Conceptualization:** Sahra Maria Anna Bucher.

**Supervision:** Patrick E. Beeler.

**Validation:** Anne Marie Schumacher Dimech.

**Writing – original draft:** Sahra Maria Anna Bucher.

**Writing – review & editing:** Anne Marie Schumacher Dimech, Beat Müller.

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
