## [Decision Letter · Decision Letter 0]

4 Mar 2024

PONE-D-23-42337Interprofessional collaboration during a specialised mobile palliative care service pilot in the rural area of LucernePLOS ONE

Dear Dr. Bucher,

Thank you for submitting your manuscript to PLOS ONE. After careful consideration, we feel that it has merit but does not fully meet PLOS ONE’s publication criteria as it currently stands. Therefore, we invite you to submit a revised version of the manuscript that addresses the points raised during the review process.

We look forward to receiving your revised manuscript.

Kind regards,

Martin Schneider

Academic Editor

PLOS ONE

Journal Requirements:

Additional Editor Comments:

The reviewers reach very different conclusions.

In addition to the comments made by reviewer #2, I would raise the following points:

• There are too many abbreviations, some are used without correct introduction. Please reduce the number of abbreviations and abstain from those that are often used for other terms (like IPC – infection prevention and control).

• One should discuss the reasons why the interviewer proceeded “without further training in conducting interviews” (page 5). This weakness would also merit a brief discussion.

Please note that if you revise the manuscript, its publication cannot be taken for granted.

Reviewers' comments:

Reviewer's Responses to Questions

**Comments to the Author**

1. Is the manuscript technically sound, and do the data support the conclusions?

Reviewer #1: Yes

Reviewer #2: Partly

2. Has the statistical analysis been performed appropriately and rigorously? 

Reviewer #1: Yes

Reviewer #2: No

3. Have the authors made all data underlying the findings in their manuscript fully available?

Reviewer #1: Yes

Reviewer #2: Yes

4. Is the manuscript presented in an intelligible fashion and written in standard English?

Reviewer #1: Yes

Reviewer #2: No

5. Review Comments to the Author

Reviewer #1: The authors present a relevant and interesting evaluation of facilitators and barriers of interprofessional collaboration in specialist palliative care. The data is relevant for health-care professionals and policy makers. The mixed-method apporach is sound and the manuscript clearly structured and reasonable to follow. The appendix does not leave any open questions. Tables and figures are presented in good quality and add to understanding of the result. Orthography and semantics are impaccable, but please note, that I am not a native speaker. In the discussion, you may consider adding some information about the evidence base of specialist palliative care, but this must not be considered mandatory. If you do consider this recommencation hellpful, you may refer to https://pubmed.ncbi.nlm.nih.gov/28676557/

Overall, we need more of such elaborate trials as presented by the working group from Lucerne.

Good job!

Reviewer #2: Researchers investigated the barriers and facilitators to interprofessional collaboration in mobile palliative care team. Whereas the topic is of great interest, the study presents multiple critical points.

Authors describe a mix method study investigating nurses’ and physicians’ experiences of collaborating in a pilot project aimed to implement home care specialised palliative care.

As general reflections, authors should better describe the pilot project with major points that will enable the readers to understand how the mobile team was formed, which professionals are part of this team and which model of care was used.

Authors should precisely state a research question.

For the qualitative part, given the small area in which the research was conducted, and the details provided about participants in the study, authors should reassure the reviewers about the respect of the anonymity of participants or provide less identifying details.

- Concerning the selection of participants authors should mention inclusion and exclusion criteria of participants and, given the aim to investigate interprofessional collaboration, precisely mention what were the professions of the “professional groups ( PG) “. Were all part of the palliative care specialised team? Were they part of the home care primary nurses, GPs or others? It needs a clear description of the population interviewed.

- Why were only professionals of the specialised mobile team invited to participate in the quantitative part of the study?

- Given that Reeves’ theory was used to analyse the data, a brief explanation of why this theory was chosen and why only two factors of this theory were used in a separate set of interviews should be provided.

- Why was content analysis chosen as a method to analyse interviews? How does content analysis link with a coding framework deductively chosen?

- The results section is divided in paragraphs with titles. How are the titles linked to the codes that emerged from the content analysis? The paragraphs are mostly descriptive of the collected data. There doesn’t appear to have a link with a coding, and a higher conceptualisation of data is needed, and a clear link to Reeves’s four factors should emerge.

- The number of participants in the survey is too low (17 participants) to contribute substantially to the understanding of the qualitative part nor allows to derive statistically substantiated data.

- Was the qualitative part conducted before or after the survey? Why was this approach chosen?

- Authors state that the results section for the qualitive part shows “relevant results”. Do they mean main themes or domains?

- The discussion section presents the facilitators and barriers of IPC. Their identification and analysis were stated as the aim of the research, for such reason should be considered as the results of the study.

- Data transcripts and coding book should not be provided for publication and interview grid and all study material presented in German should be selected and that relevant for publication should be translated into English. Quotes should be selected to represent the variety of opinions and translated into English.

Whereas an interesting area of research, the paper needs more work in its overall structure and the quality of academic English needs improvement.

6. PLOS authors have the option to publish the peer review history of their article (what does this mean?). If published, this will include your full peer review and any attached files.

Reviewer #1: **Yes: **Prof. Dr. Jan Gärtner

Reviewer #2: No

---

## [Author Response · Author response to Decision Letter 0]

1 May 2024

We would like to express our sincere gratitude for your review and constructive feedback. Enclosed, you will find the revised documents reflecting your suggestions. Direct revisions can be found in the document titled "revised manuscript with tracked changes" and "response to reviewers". We greatly appreciate your cooperation throughout this process.

---

## [Editor Report · Decision Letter 1]

5 Jun 2024

PONE-D-23-42337R1Interprofessional collaboration during a specialised mobile palliative care service pilot in the rural area of LucernePLOS ONE

Dear Dr. Bucher,

Thank you for submitting your manuscript to PLOS ONE. After careful consideration, we feel that it has merit but does not fully meet PLOS ONE’s publication criteria as it currently stands. Therefore, we invite you to submit a revised version of the manuscript that addresses the points raised during the review process.

We look forward to receiving your revised manuscript.

Kind regards,

Martin Schneider

Academic Editor

PLOS ONE

Journal Requirements:

Additional Editor Comments:

For a future revision, please set your word processor to indicate changes in the text only, not in formatting. This will make it easier to read.

Please consistently spell -ise or -ize. It is Advance (not advanced) care planning, “and” instead of “&.”

Why was there no calculation for the sample size (2.3.2 Quantitative data collection)?

Why differences in interviews with physicians and nurses (2.3.1 Qualitative data collection)?

If you manage to shorten the manuscript a little more, your readers will appreciate.

---

## [Author Response · Author response to Decision Letter 1]

17 Jul 2024

Thank you very much for your valuable feedback. We appreciate the collaboration. The adjustments can be found in the documents "Revised Manuscript with Tracked Changes" and "Answers to Academic Editor". Should you have any questions or concerns, we look forward to your feedback.

---

## [Editor Report · Decision Letter 2]

22 Jul 2024

Interprofessional collaboration during a specialised mobile palliative care service pilot in the rural area of Lucerne

PONE-D-23-42337R2

Dear Dr. Bucher,

We’re pleased to inform you that your manuscript has been judged scientifically suitable for publication and will be formally accepted for publication once it meets all outstanding technical requirements.

Kind regards,

Martin Schneider

Academic Editor

PLOS ONE
---

## [Editor Report · Acceptance letter]

2 Aug 2024

PONE-D-23-42337R2 

PLOS ONE

Dear Dr. Bucher, 

I'm pleased to inform you that your manuscript has been deemed suitable for publication in PLOS ONE. Congratulations! Your manuscript is now being handed over to our production team.

Kind regards, 

on behalf of

Dr. Martin Schneider 

Academic Editor

PLOS ONE